

# A computer architecture based on disruptive information technologies for drug management in hospitals

Ricardo Chalmeta[1], Andres Navarro-Ruiz[2] and Leticia Soriano-Irigaray[2]

[1] Grupo de Integración y Re-Ingeniería de sistemas, Departamento de Lenguajes y sistemas Informáticos, Universitat Jaume I de Castellón, Castellón, Spain
[2] Hospital General Universitario de Elche, Elche, Spain

## ABSTRACT

The drug management currently carried out in hospitals is inadequate due to several factors, such as processes carried out manually, the lack of visibility of the hospital supply chain, the lack of standardized identification of medicines, inefficient stock management, an inability to follow the traceability of medicines, and poor data exploitation. Disruptive information technologies could be used to develop and implement a drug management system in hospitals that is innovative in all its phases and allows these problems to be overcome. However, there are no examples in the literature that show how these technologies can be used and combined for efficient drug management in hospitals. To help solve this research gap in the literature, this article proposes a computer architecture for the whole drug management process in hospitals that uses and combines different disruptive computer technologies such as blockchain, radio frequency identification (RFID), quick response code (QR), Internet of Things (IoT), artificial intelligence and big data, for data capture, data storage and data exploitation throughout the whole drug management process, from the moment the drug enters the hospital until it is dispensed and eliminated.

## INTRODUCTION

Patient safety has become a key priority for the main international and national health organizations (*WHO, 2023*; *AHRQ, 2023*; *IHII, 2023*; *NQF, 2023*), especially since the report (*Kohn, Corrigan & Donaldson, 2000*) where it was identified that care errors in hospitals caused between 44,000 and 98,000 deaths per year. This report was decisive for the prevention of harms derived from health care, since it brought to light the important care and economic repercussions of clinical errors. In addition, it addressed the medication errors (ME) associated with the use of medications, considering that they were the most prevalent types of errors (*Otero López, 2010*).

Currently, the most widely accepted definition of medication error (ME) is the one given by the national coordinating council for medication error reporting and prevention (*NCCMERP, 1998*). This body defined ME as any preventable incident that can cause harm to the patient or lead to inappropriate use of medicines, when these are under the control of health professionals or the patient. These incidents may be related to professional

Corresponding author
Ricardo Chalmeta, rchalmet@uji.es

practices, products, procedures or systems, including failures in the prescription, communication, labelling, packaging, naming, preparation, dispensing, distribution, administration, monitoring and use of the drugs, as well as issues involving staff training.

Examples of errors may be skipping a dose, omitting the medication, confusing one medication with another, altering the medication schedule, altering the dose, making a mistake in the route or form of administration, altering the duration of the treatment, not having the medication at the time it is to be taken, or not performing any other routine directly involved with the medication and that is essential for taking it (for example, food intake, doing some exercise, *etc.*).

Several studies have revealed the importance of ME in patients linked to health care (*Rothschild et al., 2005*). It is estimated that MEs cause adverse reactions in 1.8% of hospitalized patients, and are responsible for 4.6% of hospital admissions. These figures are higher in patients over 65 years of age, reaching 9.7% of hospital admissions. In the specific case of ME in hospitals, in a study carried out by *Otero López et al. (2003)*, it was shown that the rate of ME was 10.4%.

In order to significantly reduce ME in the hospital environment and increase patient safety, while improving efficiency in operational processes, it is necessary to improve the quality of drug management in hospitals (*Pereira et al., 2016*), throughout the entire manufacturing chain until its final disposal (*Olsen & Borit, 2013*). The implementation of an efficient drug management system in hospitals will make it possible to provide the right product, under the necessary conditions, in the expected place, to the right patient and at the right time (*Martínez Pérez, Vázquez González & Dafonte, 2016*).

However, at present, the drug management in hospitals is not adequate due to several factors, such as manual procedures, the lack of visibility of the hospital supply chain, the lack of standardized identification of medicines, inefficient stock management, an inability to follow the traceability of medicines, and poor data exploitation (*Haji et al., 2021*).

Disruptive information technologies such as Blockchain, Internet of Things (IoT), artificial intelligence (AI) and big data can be used to develop and implement a drug management system in hospitals that is innovative in all its phases, that is, reception, storage, handling, processing, dispensing, administration, and recycling of waste (*Hussien et al., 2021*). The combination and use of these technologies can guarantee the capture, transparency, immutability, high availability, real-time traceability, security and analysis of the information related to drug management, which will improve the quality and safety of patient care (*Gaffney, 2018*; *Jangir et al., 2019*).

However, there are no examples in the literature that show how these technologies can be used and combined to overcome the existing problems and achieve efficient drug management in hospitals. Existing studies have focused on analysing the usefulness of only one or two of these technologies; for just some aspects of drug management, without considering the whole process from the moment the drug enters the hospital until it is dispensed and eliminated; and only for data capture (*Chen, Chen & Yang, 2020*; *Liu et al., 2022*), data storage (*Pandey & Litoriya, 2021*) or data exploitation (*Garcia et al., 2021*), without taking these three issues into account at the same time.

To help solve this research gap in the literature, this article proposes a computer architecture for the whole drug management process in hospitals that uses and combines different disruptive computer technologies such as Blockchain, RFID, QR codes, Internet of Things (IoT), artificial intelligence and big data, for data capture, data storage and data exploitation. Its implementation in a hospital will allow the automation of tasks; the registration of all operations carried out on medicines, thereby guaranteeing the immutability, traceability and security of the registered data and its interconnection with other systems; and the exploitation of the data to generate information to help with the management and use of medicines in hospitals.

The research method followed to develop the computer architecture was as follows. First, in accordance with *Villegas-Ch et al. (2021)*, it was important to identify the problem and the environment where the research was being conducted. The results are shown in the section "Existing problems with the traceability of medicines in hospitals". Second, the literature that deals with this line of research was analysed. The results are shown in the section "Disruptive information technologies for drug management: related work". Third, the computer architecture was developed based on the authors' experience both in the application of disruptive information technologies to different scenarios and in drug management in hospitals. The architecture is shown in the section "Proposed computer architecture". Four, according to *Dzwigol (2020)*, the validation of the computer architecture was conducted through experts' opinion. An explanation of how the validation was conducted is shown in the section "Validation".

The rest of the article discusses the findings and novelty of the proposal, and shows the conclusions, including a discussion of future research and limitations.

## EXISTING PROBLEMS WITH THE TRACEABILITY OF MEDICINES IN HOSPITALS

The intra-hospital drug process begins with the management of the drug supply and their reception by the hospital's pharmacy service. Subsequently, after the medical prescription, the drug is distributed and dispensed, which may entail prior handling, preparation or reconditioning by the pharmacy service. The drugs are dispensed to the different hospitalization units for their administration by the nursing staff (Fig. 1).

To ensure the quality and safety of patients' care, it is necessary to carry out a correct drug management, something that is also required by the new regulations currently in force in different countries such as those belonging to the European union (EU), United States, *etc*. However, the real situation is that the drug management of the intra-hospital drug chain is inefficient, with a lack of transparency in some processes and not completely safe in terms of administration to the patient (*Raijada et al., 2021*). The main problems existing at present are the following:

**1. *Inadequate presentation format of the data to be recorded.*** Currently the data required according to the regulations are the lot number and the expiration date of the drug, which must appear incorporated in the secondary package (outer package) (*Felix et al., 2019*). However, this presentation format does not correspond to current needs (*Küng et al., 2021*; *Hsieh et al., 2021*). This secondary package is discarded by hospital

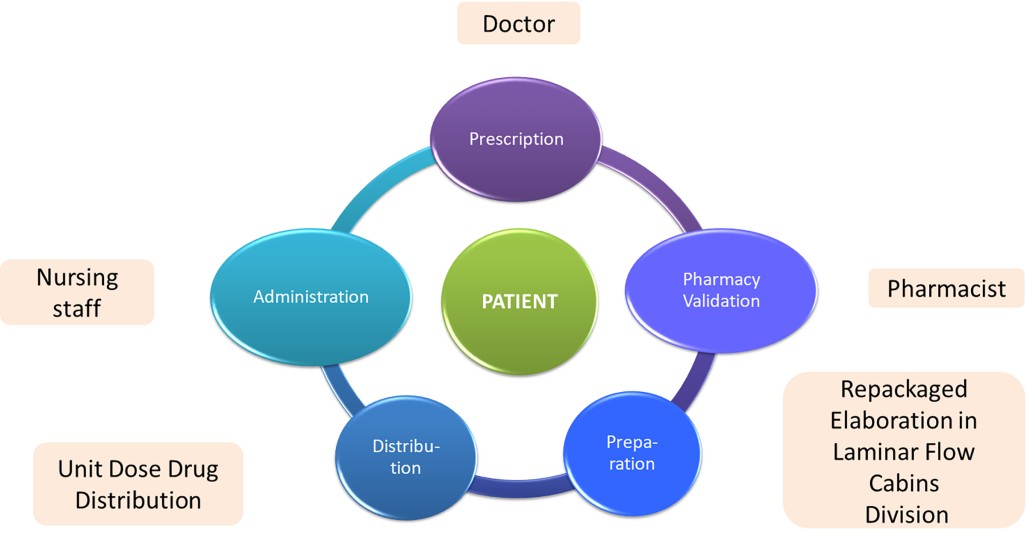

**Figure 1 Activities and human resources involved in the drug management process in hospitals.**

pharmacy services due to lack of identification of the individual doses, and the content is extracted either for processing or for conversion into unit doses.

Next, hospital pharmacy services generate a label with the name of the medication, active ingredient, expiration date, and lot number. This is due to many drugs cannot be administered directly—they need subsequent preparation. For example, in the case of sterile preparations, the presence of materials that release particles in pharmacy clean rooms must be avoided, so drugs must enter without the secondary packaging (*Casaus, 2011*). Therefore, these codes would also have to appear on the primary packaging (the container or any other form of packaging that is in direct contact with the medicine), and thus facilitate traceability in the very moment the medicines are prepared.

**2. *Current intra-hospital drug management processes must be adapted to the regulatory changes.*** As a result of the alarming increase in counterfeit medicines detected, different countries are adopting new regulatory procedures. For example, the European Union directive 2011/62/EU (*EU, 2011*) establishes that security features on the outer packaging of pharmaceuticals should be harmonized in all EU countries to help the identification and verification of the authenticity of the drugs.

Along with this, the *EU (2016)* establishes that manufacturers must encode the security identifier in a two-dimensional barcode, readable by optical technology, which will allow identification of the name, lot number and expiration date of the drug, among other elements. These new regulations will improve drug traceability and management, forcing changes in the business process and hospital computer systems.

**3. *Inefficient data capture procedure.*** Data regarding the drug lot number and expiration date are recorded manually, although in practice not for all the drugs that are administered but only for those that are legally required to be registered, such as blood products or biological drugs. This is due to the large volume of drugs that are handled daily by hospital pharmacy services, since because it is a manual process, it is impossible to

record the lot number and expiration date of all drugs (*Fan et al., 2022*; *Barakat & Franklin, 2020*; *Sessions et al., 2019*).

**4. *Incomplete data record.*** The implementation of traceability systems must make it possible to completely reproduce the drug route in the hospital. Consequently, recording the drug lot number and the expiration date is not enough to establish an adequate traceability system (*Grau et al., 2018*), since it is necessary to know other data such as where the medicine is located, when it was sent from the pharmacy service, when it was administered to the patient, who has been in charge of each task, if the drug has been stored properly, the speed of administration, in which infusion pump it has been administered, *etc.* (*Martínez Pérez, Dafonte & Gómez, 2018*).

Traceability must be established both at the drug level (high-risk medication, blood products, advanced therapies, *etc.*), as well as at the processes level (drug reconditioning, dose individualization, manufacturing of master formulas and medicines, *etc.*) and results levels (pharmaceutical alerts, drug surveillance, drug effectiveness, *etc.*) (*Chien et al., 2021*; *Lawal et al., 2020*).

**5. *Little analysis of the results.*** Traceability is incomplete if it does not include an analysis of the results in the field of pharmacotherapy, since an adequate record of the data would allow monitoring of the patients and analysis of the achievement of health results that are pursued considering different variables (age, sex, physical condition, previous pathologies) (*Kapetaneas & Kitsios, 2022*; *El Morr & Ali-Hassan, 2019*).

Therefore, from a diagnosis of the current situation, it can be concluded that it is unfeasible to adequately control the traceability of medicines in the different stages of the hospital drug process, from their prescription to their administration to the patient. This makes it impossible to ensure, for example, that the product supplied matches the medical prescription, that it has been kept in optimal conditions until its administration, or that it reaches the correct place at the correct time and is given to the correct patient, which are aspects that are crucial to obtain the greatest safety for the patient at all times, and to achieve maximum operational efficiency. Therefore, it is necessary to improve the traceability of pharmacological treatments carried out in hospitals (*Leng, Tan & Wang, 2021*).

# DISRUPTIVE INFORMATION TECHNOLOGIES FOR DRUG MANAGEMENT: RELATED WORK

## Drug identification and data capture

Medicines are currently identified with a barcode on their outer packaging. Barcodes have been widely used to improve the safety and efficiency of drug storage and preparation in hospitals (*Sriboonruang & Rattanamahattan, 2022*). In addition to this, a new use of barcodes is starting to be applied in hospitals. The objective is to dispense the correct medication at the right time and to the right patient, which is known as barcode medication administration (BCMA). In this case, it is necessary to scan a patient's unique barcode and medication barcodes before administering a dose in order to verify that both of them are correct (*Barakat & Franklin, 2020*).

A barcode is a pattern consisting of white and black stripes. It only records horizontal data and is one-dimensional (*Jessurun et al., 2022*). The QR code is an upgraded version of the barcode that records both vertical and horizontal data. The barcode represents a number, while the QR has a string of characters and can be generated redundant to resist physical deterioration. QR codes are therefore better than barcodes. In both cases, the functionality is the same. Through optical sensors, scanners detect different reflected light for identification. The reader consists of a camera and a processor, a mobile phone being the most intuitive example (*Pruitt et al., 2023*).

As they are simple "drawings", their cost is extremely low and they have other advantages such as high reliability and fast input speed. However, they also have some shortcomings. They must be read optically, so their size has to be adapted to the characteristics of the reader. Reading and processing is very fast, but it requires an optical focus with the barcode perfectly aligned with the scanner in order to be effective. Therefore, individual reading is common. Finally, neither barcodes nor QR is capable of carrying out tracing and real-time tracking of the source of products (*Liu et al., 2022*).

In addition to barcodes and QR, there are two other possible ways to record more information about the drug (both the drug sent by the supplier and the drug resulting from the processing carried out by the hospital pharmacy service) and to read it: RFID and IoT devices.

***Using RFID UHF radio frequency identification.*** In this case, a passive device (called a tag) would be assigned in the form of a small very economical label that costs just a few US cents (*Ramos et al., 2022*). Each tag can carry a different identification number and can store different information such as the name of the drug, dosage, raw materials, efficacy, production date, expiration date and manufacturer (*Islam & Islam, 2022*). The tag can be read remotely with an antenna and a receiver.

A drawback of ultra high frequency (UHF) RFID is that the cost of the reader rises considerably as the distance between the antenna and the tags increases (hundreds of US dollars). An alternative to this is to use near field communication (NFC) technology, which is a variant of passive RFID, where there is a short distance between the tag and the antenna. In this case, the tag is active, since it receives enough energy by induction to start up a programmable processor (*Balasubramanian, Vivekanandhan & Mahadevan, 2022*). The cost of NFC tags has decreased considerably with their widespread diffusion and the readers have also become cheaper, with a price that currently stands at around USD 20. The drawback of this technology is that the interaction between tag and reader is designed to be carried out one by one and is not as fast as UHF passive RFID (*Li et al., 2020b; Catarinucci et al., 2012*). The best-known examples are "contactless" bank cards.

The use of RFID in hospitals could speed up drug management and effectively prevent counterfeit, missed or expired drugs and the occurrence of errors (*Chen, Chen & Yang, 2020*). RFID can be used for four functions in hospitals: tracking, alarms, automation and management.

Tracking: RFID can be used to track the movement of people or objects (*Ruan et al., 2018*). Examples include tracking the real-time location of individuals with dementia; locating patients and staff in an emergency; tracking infected patients during the

COVID-19 pandemia (*Camacho-Cogollo, Bone & Iadanza, 2020*) or locating moving objects such as syringes, beds or surgical tools (*Lee et al., 2019*).

Alarms: RFID tags can contain different patient information such as the patient's ID, name, electronic health record, allergens, *etc.* Therefore, RFID labels allow patient identification and verification and can be used to generate alarms and indications to keep patients safe and avert potentially life-threatening circumstances. Examples would be cases in which hospital staff use or process the device in an incorrect way (*Haddara & Staaby, 2018*) or as a means to avoid medical errors such as prescribing the wrong medicine to the wrong person (*Ebrahimzadeh et al., 2021*) or even operating on the wrong patient (*Li et al., 2020a*, *2020b*).

Management: RFID allows automatic data collection, thus avoiding the need for medical and management staff to spend so much time on administrative tasks (*Vagaš et al., 2019*). For example, *Ko & Woo (2018)* proposed an RFID UHF system for drug runout detection. Another example of using RFID for management in hospitals is to use information on patients' location to discover bottlenecks.

Automation: RFID allows manual activities to be automated. For example, if each container of a medicine carried an RFID tag, when the medicines sent by the supplier arrived at the hospital pharmacy service, it would suffice to pass the entire package close to an antenna to read all the tags at once and it would thus be possible to verify that the order is correct. A conceptual model to do this has been proposed by *Liu et al. (2022)*.

In addition, it is important to remark that there are several technical, economic and social barriers that hinder the implementation and adoption of RFID in hospitals. Technical barriers can be summarized as system faults, lack of interoperability with other health information systems, misplaced or destroyed tags, problems with the readers, and interferences with other medical devices (*Abugabah, Nizamuddin & Abuqabbeh, 2020*). The economic barriers are mainly the high cost due to equipment and software acquisitions and maintenance as well as personnel training and education. Finally, social barriers refer to security concerns about the privacy of sensitive personal and health data, and the lack of worldwide RFID technology regulations and guidelines (*Zhang, Fu & Li, 2019*; *Vagaš et al., 2019*).

*Using IoT devices.* IoT devices have processing capacity and are autonomous in terms of connection (*Ali, Ishak & Bhatti, 2021*). Therefore, they can communicate *via* the Internet without requiring any human interaction (*Kumar, Tiwari & Zymbler, 2020*). IoT devices are being used in hospitals for five purposes (*Sharma, Kaur & Singh, 2021*):

1) For remote patient monitoring. For example, *Buthelez et al. (2022)* developed an IoT-based system which monitors the heartbeat of remote patients; *Chang et al. (2021)* developed an IoT real-time monitoring and nursing intervention for patients with insomnia; and *Latif et al. (2020)* proposed an IoT prototype of Wireless Sensor Network and Cloud based system to provide continuous monitoring of a patient's health status.

2) For medical alert systems. For example, *Kotb et al. (2022)* used an IoT-enabled miniaturized remote auto-injector supported by smartwatch health monitoring for the emergency medical treatment of allergy-induced anaphylaxis patients; and *Wang et al.*

*(2019)* developed a wearable device for monitoring blood pressure, which analyses in real time and predicts the occurrence of strokes through machine learning algorithms.

3) For hospital management. For example, *Xu et al. (2020)* applied an IoT management system for hand hygiene and proved that there was a significant improvement in the hand hygiene compliance rates among the members of the medical staff before and after contact with patients and their surrounding environment.

4) For continuously monitoring patients with chronic diseases. For example, *Wang & Song (2023)* proposed an edge-assisted Internet of Medical Things-based smart-home monitoring system for the elderly with chronic diseases.

5) For surgical robotics that offer more precise results than human doctors, in orthopaedic trauma, for example (*Merle et al., 2022*).

However, the use of IoT for drug management in hospitals is scarce. Although some studies related to drug management and IoT have been conducted, they are focused on activities outside the hospital (*Cocian, Morales & Schneider, 2023*). Such applications are mainly concerned with tracking the logistics of medical products (*Nanda, Panda & Dash, 2023*) and remote monitoring of medication behaviours of patients once they have left the hospital (*Roh et al., 2021*).

As a result of this comparative analysis for drug management in hospitals, it can be concluded that IoT is not eligible for use as a drug package due to the size of the devices required and because, at present, its price may make it unfeasible. Indeed, an IoT device could be more expensive than the drug itself. On the other hand, UHF RFID labelling is very useful due the speed at which sets of tags can be read compared to current drug barcode scanning (*Petro et al., 2009*; *Iadanza, 2012*; *Wu, Kuo & Liu, 2005*; *Bevilacqua et al., 2013*), but readers are expensive. Finally, barcodes and QR codes are easy and cheap to generate, but their reading requires direct optical focus, which can be a serious practical drawback.

In all cases it is important to consider the size. Labelling a medicine box or bottle is not the same as labelling a single-dose blister pack. Sustainability issues must be borne in mind too. The carbon footprint generated by labelling is also important (*Kapoor, Zhou & Piramuthu, 2009*). A drug packaged in a single dose and labelled with an RFID must be discarded or reused after its administration. Reuse is a difficult process due to the inconvenience in handling the label, so recycling seems the only way, but the cost of the labels is not recovered. To address this, ink-printed disposable RFID tags have been proposed (*Liu et al., 2022*), but they are not known to be commercially available. On the other hand, barcodes/QR codes printed on article are easily recyclable with the standard recycling of article. Table 1 summarizes the advantages and disadvantages of barcode/QR, RFID and IoT for drug management in hospitals.

## Data storage

Blockchain technology can help solve the problem of incomplete data recording, since it allows the traceability of medicines to be managed from the moment medicines arrive at the hospital until they are dispensed to the patient (*Kim et al., 2019*). It can also help

**Table 1 Advantages and disadvantages of barcode/QR, RFID and IoT for drug management in hospitals.**

| | Advantages | Disadvantages |
|---|---|---|
| Barcode/QR | • Low cost<br>• High reliability<br>• Fast input speed<br>• Easily recyclable | • Requires optical focus with a perfect alignment<br>• Individual reading<br>• Cannot perform tracing and real-time tracking |
| RFID | • Many measurements at the same time<br>• A direct line of sight is not needed to read the tag information<br>• Tracking and tracing are better than with barcode<br>• Allows real-time response and end-to-end visibility<br>• Less error in patient identification<br>• Allows medical reporting<br>• Improved worker productivity<br>• Automatic data capture<br>• Improvement of resource allocation and utilization<br>• Real inventory level | • High cost of implementation<br>• Resistance to change (RFID technologies will have a significant impact on hospital workflow and business processes)<br>• Lack of interoperability with the existing IT infrastructure<br>• Technical limitations, interferences, system faults, *etc.*<br>• There are no industry standards or guidelines to guide implementations<br>• Data security and privacy problems<br>• Reuse and recycling difficulties |
| IoT | In addition to RFID advantages:<br>• Processing capacity<br>• Autonomy in the connection | In addition to RFID disadvantages:<br>• Higher cost<br>• Large size |

guarantee their quality and prevent counterfeits from reaching the end user (*Gaffney, 2018*; *Sylim et al., 2018*; *Omidian & Omidi, 2022*; *Pandey & Litoriya, 2021*).

In this way, with blockchain-based software, it would be possible to collect and store for each clinical diagnosis, among other data, the doses prescribed and those actually administered to the patients with all the data referring to the characteristics of the drug used (name, lot number, expiration date, history, location, trajectory, *etc.*), the characteristics of the patients (sex, age, pathologies, *etc.*), the effects of the therapy, the actions of the professionals in each part of the process, and the material and the equipment used, all of which will ensure the effectiveness, safety, efficiency and quality of the process (*Odeh, Keshta & Al-Haija, 2022*).

Blockchain is a protocol for the communication and storage of information in a distributed database system (*Lu, 2022*). It consists of a network of nodes where all its participants can access a copy of the database, thus eliminating the figure of an intermediary node that centralizes storage and distribution and, at the same time, making it possible to verify and approve transactions between participants.

The operating procedure implies that once a transaction related to the drug is registered, it cannot be modified. This feature, immutability, is crucial to ensure data integrity and security (*Garcia et al., 2022*). The combination of these two characteristics makes the system transparent and secure for its participants, who can access and consult the records

at any time, with the confidence that they are reliable records that cannot be altered over time, and the certainty that the other participants in the operation have exactly the same information (*Jamil et al., 2019*).

Table 2 presents the main studies related to blockchain and drug management as well as their limitations regarding the contributions they can make to drug management in hospitals. All these studies highlight the advantages of using blockchain for healthcare data management. Although they focus on drug data management, only one considers activities related to drug storage, management and dispensing inside the hospital. The other studies are focused on the supply chain, mainly with the aim of preventing drug counterfeiting.

## Data exploitation

Data by itself does not offer any value but must be processed to generate information that can help make better and faster decisions. A Hospital Business Intelligence tool can be developed using the data generated and recorded throughout the drug management process in hospitals. This tool can be used for two purposes:

– *Improve the management of hospitals and health centres*. The information obtained will make it possible to measure the efficiency of each of the services and to anticipate the demand for them, the availability of professionals (*Basile et al., 2022*) or to detect bottlenecks (*Martínez Pérez, Dafonte & Gómez, 2018*). Therefore, actions can be undertaken to improve weak points detected, and best work practices (who has to carry out each task and when) will be developed. Moreover, maximum efficiency and effectiveness in the allocation of available human and technological resources will be achieved in the drug circuit shown in Fig. 1.

– *Evaluate the drug results in patients considering different variables*. This will make it possible to (1) establish behaviour patterns in relation to the effects of drugs in the different types of patients and diseases; (2) improve medical decisions; (3) develop new drugs that are cheaper and more effective; and (4) analyse, thanks to traceability, the possible causes of deviations from the expected effect in a patient after the administration of a drug (*Raza et al., 2022*).

The exploitation of health data is a major stream of current public policy. For example, the European Union has launched different funding programmes to support research and innovation projects to boost real-world health data integration, sharing, exploitation and use (*EU, 2023*). However, current research about data exploitation related to drug management in hospitals is scarce. Table 3 shows the journal articles about data exploitation for business intelligence in hospitals using drug data found in the Web of Science following a search with the keywords "hospital" and "business intelligence" and "drug".

## PROPOSED COMPUTER ARCHITECTURE

There are no examples in the literature that show how the above technologies can be used and combined for efficient drug management in hospitals that overcomes the existing problems. To solve this research gap, the authors have developed a computer architecture

**Table 2 Contributions and limitations of the studies related to blockchain and drug management in hospitals.**

| Study | Contribution | Limitation |
|---|---|---|
| Sylim et al. (2018) | Proposed a model for drug counterfeit control using blockchain, barcode and RFID | It is focused on activities performed outside the hospital |
| Fan et al. (2018) | Proposed a blockchain-based information management system to handle patients' information. Shows how to combine blockchain with security and authentication | Drug traceability inside the hospital is not considered |
| Kim et al. (2019) | Developed a Patient-centric medication history recording system using blockchain. It allows data sharing among different hospitals thereby preventing patients from recording the prescription information by themselves | It is an experimental proof of concept validated in the laboratory. Findings should be checked in a real environment |
| Jamil et al. (2019) | Proposed a manager of histories, treatments and recipes using Hyperledger Fabric. It enables doctors, nurses, patients and pharmacists to manage, access and share personal medical records and a complete individual drug life cycle in a secure and accountable way | It is an experimental proof of concept validated in the laboratory. Findings should be checked by increasing the network size and by deploying the system in a real environment |
| Jangir et al. (2019) | Proposed a pharmaceutical supply chain management framework based on blockchain. Proved the traceability and immutability of blockchain. Developed a case study on a public Ethereum network | It is focused on activities performed outside the hospital |
| Yazdinejad et al. (2020) | Proposed a decentralized authentication of patients in a distributed hospital network, by leveraging blockchain. This proposal makes it possible to increase throughput, reduce overheads, improve response time and decrease energy consumption in the network | It is not focused on drug management in hospitals |
| Kumiawan, Kim & Ju (2020) | Proposed a blockchain-based medicine supply chain management to prevent the presence of counterfeit medications | It is focused on activities performed outside the hospital |
| Li et al. (2021) | Provided use cases of blockchain-enabled IoT for fighting against the COVID-19 pandemic, including the prevention of infectious diseases, location sharing and contact tracing, as well as the supply chain of injectable medicines | It is focused on activities performed outside the hospital |
| Wang et al. (2021) | Analysed the risks that impact the reliability of blockchain to prevent the presence of counterfeit medicines | It is focused on activities performed outside the hospitals |
| Garcia et al. (2021) | Demonstrated that smart contracts for electronic prescription can be implemented on a Byzantine Fault Tolerant-based platform. The proposal can help to prevent patient misuse of medications and make it hard to change confidential data to obtain illegal benefits | It is an experimental proof of concept validated in the laboratory. Findings should be checked in a real environment. Activities related to drug storage, management, and dispensing inside the hospital are not considered |
| Sharma, Kaur & Singh (2021) | Identified the main applications of blockchain in hospitals: data storage, data sharing, drug traceability, clinical trials and remote patient monitoring | There is no guide, methodology or model about how to develop a blockchain for these applications |
| Hussien et al. (2021) | Highlighted the reasons for implementing blockchain technology in the healthcare industry | There is no guide, methodology or model about how develop a blockchain for drug management |
| Uddin et al. (2021) | Presented an overview of product traceability issues in the pharmaceutical supply chain and envisaged how blockchain technology can provide effective provenance, track and trace solutions to mitigate counterfeit medications. Offered a good comparison of Hyperledger Fabric and Hyperledger Besu | It is focused on activities performed outside the hospital |
| Pandey & Litoriya (2021) | Proposed a resilient electronic health network using blockchain to tackle the problem of counterfeit medicine. This solution is based on recording the medicine logistics requirements from the manufacture of the medicine to the patient on the blockchain network | It is an experimental proof of concept validated in the laboratory. Findings should be checked in a real environment. Activities related to drug storage, management and dispensing inside the hospital are not considered |

(Continued)

| Study | Contribution | Limitation |
|---|---|---|
| *Garcia et al. (2022)* | Developed a decentralized data governance framework for e-prescription. Proved the immutability of smart contracts and analysed the execution time of TenderMint and HL Fabric | It is not focused on drug management in hospitals |
| *Odeh, Keshta & Al-Haija (2022)* | Investigated how blockchain technology can be applied to improve the overall performance of the healthcare sector | It is not focused on drug management in hospitals |
| *Omidian & Omidi (2022)* | Analysed the projects of the U.S. Food and Drug Administration in the fight against counterfeiting, using blockchain and QR | It is not focused on drug management in hospitals |

**Table 3 Contributions of the studies about data exploitation for business intelligence in hospitals using drug data.**

| Year | Contribution |
|---|---|
| *Li et al. (2019)* | Proposed a real-time social media analytics framework to create knowledge about adverse drug reaction surveillance |
| *Hajjami, Berrada & Harti (2020)* | Obtained knowledge from the feelings and opinions that patients express on the web and social networks |
| *Masuda et al. (2021)* | Proposed an enterprise architecture for new drug discovery to be implemented in the pharmaceutical industry |
| *Galli et al. (2021)* | Proposed machine learning coupled with stochastic optimization to analyse historical drug usage patterns to minimize inventory levels as well as the need for emergency replenishments |
| *González-Pérez et al. (2022)* | Developed Business Intelligence for the visualization and data analysis of Telepharmacy activity indicators on a hospital pharmacy service scorecard |
| *Xia (2022)* | Developed a deep-learning based approach to integrating historical profiles for the detection of adverse drug events |
| *Raza et al. (2022)* | Analysed the biggest breakthroughs for drug discovery, dosage form design, polypharmacology, and hospital pharmacy using AI |
| *Sajogo, Teoh & Lebedevs (2023)* | Developed a tool for the documentation and analysis of clinical interventions to support medical and nursing education |
| *Ziaee, Shee & Sohal (2023)* | Improved the Australian pharmaceutical supply chain using Big Data |

that combines QR, RFID, Blockchain, IoT, AI and big data to support drug management in hospitals. The proposed computer architecture is organized on three levels: drug identification and data capture through IoT, QR and RFID; data storage through Blockchain; and data exploitation through a business intelligence system (Fig. 2).

## Drug identification and data capture

The computer architecture proposed for reading data: (1) Uses IoT in those phases of the process where information needs to be constantly monitored, such as temperature control in the warehouse, and (2) Proposes two alternatives to record the information about the drugs and the tasks performed with them:

– To achieve the maximum agility in the process, the best solution is to use RFID. This solution requires a higher initial investment since it is necessary to buy and install the antennas, but it is quickly amortized because of the automation of the data capture for drug traceability.

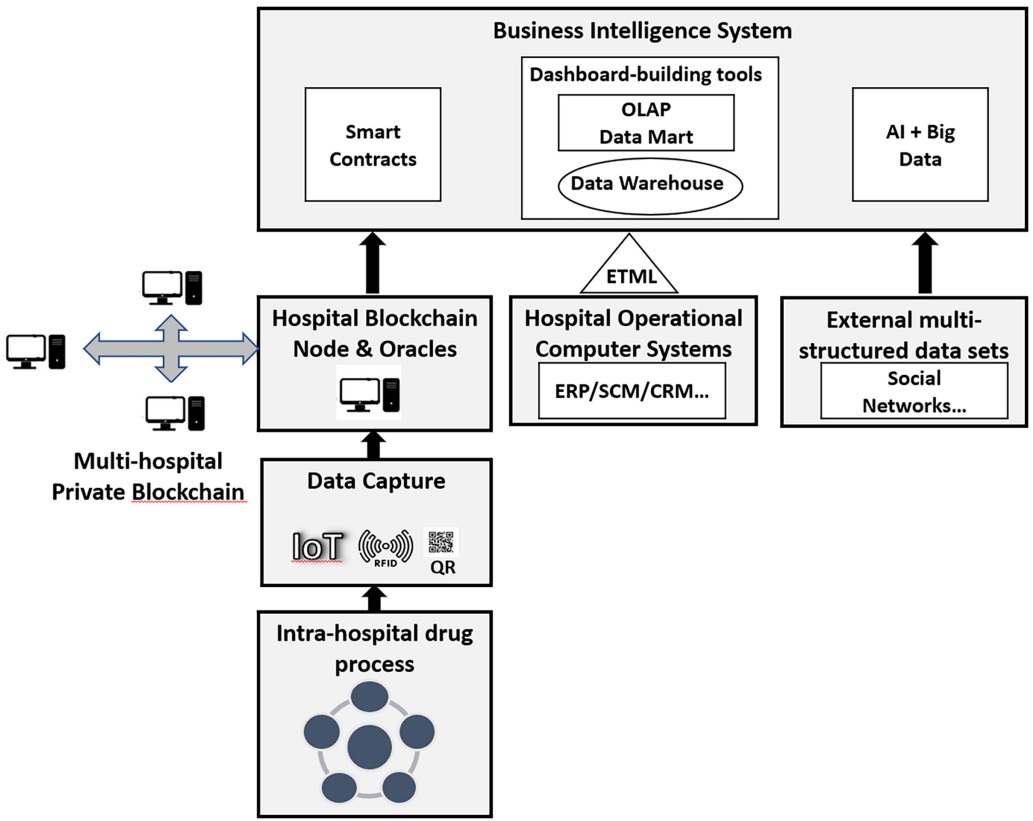

**Figure 2 Computer architecture that combines QR, RFID, blockchain, IoT, AI and big data for drug management in hospitals.**

– To prioritize minimum environmental impact, the best solution is to use QR codes, since the label is recyclable, although the registration process will be manual, and therefore it will be slower and more cumbersome than with RFID.

To speed up the process, in the tasks where there is a high volume of data to be recorded, such as when a set of medicines sent by the supplier arrives at the hospital pharmacy service, an industrial QR/barcode reader can be used. This device can read a set of codes at the same time, although reading them together may not guarantee detection of all the codes. For these devices, associated microprocessing hardware will be necessary, which will require its own programming and configuration to connect to the server.

On the other hand, in the process tasks with a smaller volume of data to be captured, such as the dispensing of medicines in each hospital room, a stand-alone barcode reader, such as a smartphone, is enough. For these devices, it will be necessary to programme and configure an app to connect them with the server.

## Data storage: blockchain-based traceability system

The blockchain for the management of medicines in hospitals should be private, and it could be based on an open source without licenses such as hyperledger fabric (*Leng, Tan & Wang, 2021*). The blockchain will have one node for each hospital that uses the

blockchain-based traceability software. This solution has been chosen due to its low cost in comparison to public blockchains and due to the privacy, it provides, which is not incompatible with its verifiable immutability.

Using a public blockchain network, such as one compatible with Ethereum, represents an unpredictable expense in network fees (called "gas") on each transaction (data storage on the blockchain). Given the high volume of daily transactions in a hospital related to drug traceability (it is estimated at a minimum of 6,000 for a medium-size hospital), the cost could be unaffordable, taking into account that the costs of the service cannot be predicted in the medium term.

There is an intermediate solution, which is to use a free immutable distributed storage solution such as IPFS and perform a single daily transaction to the public blockchain containing an immutable address that references all the data generated by the hospital during the day. The problem of how to provide privacy remains, and this would add additional complexity and greater difficulty when it comes to extracting the stored data for exploitation. To guarantee the anonymity of the data, in the event of not allowing hospitals to access patient data from other hospitals, data can be saved in pseudonomized form.

Regarding the interaction between the blockchain and the outside, this is performed through specialized programs (called oracles) that use remote procedure calls offered by the blockchain and receive data or events from the outside. For the management of medicines in hospitals, it will be necessary to programme oracles for different functions such as data collection through reading devices, mobile apps or a rest application programming interface (API).

Finally, to ensure security, the private blockchain must be protected at network level to minimize the chances of an external attack. For this, a virtual private network (VPN) must be developed in which the nodes will be located. This network can be developed using open-source software (such as OpenSSL) that is available for various systems. The VPN is deployed over the Internet, so that all the hospitals remain transparently connected, as well as any new node (new hospital) added to the blockchain at a later date. In addition, the oracles, the data acquisition devices and the API clients must also be protected by VPN. Therefore, in each hospital using a blockchain-based traceability system, a VPN will be deployed (Fig. 3).

## Data exploitation: business intelligence system

Data that is useful for drug management can be structured or unstructured. Structured data is data that is already organized and formatted and can be easily extracted through structured query language (SQL). For example, this could be the data that is stored in the hospital management information systems such as patient history, data on health personnel, *etc*. Unstructured data, on the other hand, is data that is not in a specific format, and therefore will be more difficult to analyse. Examples of unstructured data can be the data obtained from IoT sensors, messages on social networks from patients and hospital employees, emails, data from forms and surveys, *etc*.

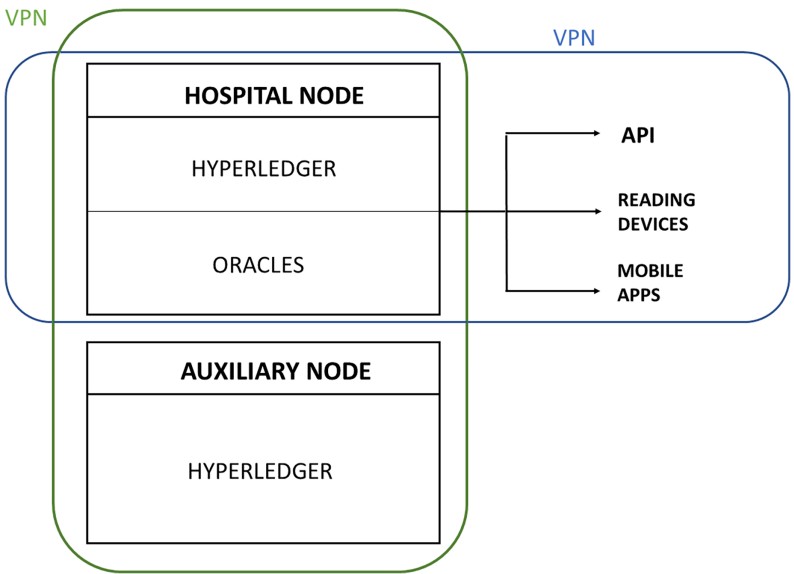

**Figure 3  Private blockchain protection through VPN to ensure security.**

For data processing to support drug management in hospitals, it is necessary to develop a third generation Business Intelligence (*Attar & Chalmeta, 2023*), which will be made up of three modules.

A module will allow the creation of intuitive drag-and-drop reports from structured data using data warehouses and dashboard-building tools. Through extraction transformation migration and loading tools (ETML), data from hospital operational computer systems is centralized in a data warehouse, clustered in data marts and processed using on-line analytical processing (OLAP) tools to calculate and display indicators in user interfaces such as dashboards, spreadsheets, *etc.*

Another module will exploit the traceability data stored in the blockchain using smart contracts (*Garcia et al., 2022*). Smart contracts allow codification and validation of all the circumstances that must be met to execute an agreement between different parties, and autonomously execute its clauses when the conditions of each of them are met, without the mediation of the participants involved or third parties. In this way, the tasks and business rules that take place in a process can be automated.

To manage the traceability of medicines in hospitals, it will be necessary to programme different smart contracts. For example, for stock management, alerts can be programmed due to storage conditions, scheduled and/or urgent orders (for example, entry from the emergency room with a prescribed drug not available in the destination unit); for the prescription system, alerts can be programmed due to pregnant and/or lactating women, drug protocols (for example, antibiotic therapy, insulin or anticoagulation), allergy and intolerance, incompatibility, or incorrect dose, route, or concentration; and for the administration system, alerts can be programmed for unscheduled hours, confirmation of extra medication, security (for example, to warn if the drug selected for administration to the patient does not coincide with the one actually prescribed), *etc.*

Finally, the third module will allow structured and unstructured data to be collected from different sources, including external hospital sources such as social networks, and applying the filters and patterns needed to obtain only the valuable data. Different advanced Big Data/AI techniques can be used with this data to inquire (query, hyperlink), interact (profiles, agents, *etc.*), investigate (term analysis, automatic query guidance, parametric search, *etc.*) and improve (machine learning, classification by sentiment analysis, feature classification, *etc.*) (*Khanra et al., 2020*; *Jarrahi, 2018*; *Orenga-Roglá & Chalmeta, 2018*).

## VALIDATION

A qualitative evaluation of the computer architecture proposed was carried out by three academics and three practitioners with extensive experience in drug management in hospitals and computer systems. The opinions of the experts were collected through individual, open and semi-structured interviews. Therefore, any response and improvisation were allowed. Their opinions were based on their experience and knowledge.

Interviews lasted around 40 min, and were carried out by one interviewer. Interviewees were interrogated about the practical utility of the computer architecture; the adequacy of the presentation and structure of the computer architecture; the completeness, intelligibility and level of detail of the computer architecture; the errors and problems encountered; and proposals for improving the computer architecture. Interviewees were also asked to identify the main strengths and weaknesses of the computer architecture, comparing it with the current computer architectures for drug management in hospitals. A report was written based on the interviewees' feedback and was analysed to improve the computer architecture with the suggestions proposed.

The interviews highlighted that the computer architecture is a useful tool for drug management in hospitals, as well as acknowledging, with minor shortcomings, the completeness, intelligibility, level of detail and correctness of the proposal.

The main strengths of the computer architecture emphasized by the experts were: the computer architecture covers the three phases of drug data management in hospitals (data capture, data storage, and data processing); the proposal of a private blockchain instead of a public blockchain; the integration of smart contracts inside the business intelligence system; and the proposal of a third generation business intelligence system to support decision making related to drugs in hospitals. On the contrary, the main shortcomings found were: the necessity of giving some examples of when to use IoT, RFID and QR codes; and the necessity of considering how to ensure data privacy. The computer architecture was modified considering these suggestions.

## DISCUSSION

Management of the intra-hospital drug chain is inefficient, with a lack of transparency in some processes and not completely safe in terms of administration to the patient (*Raijada et al., 2021*). This study has identified and collected the five main problems regarding drug management in hospitals reported in the literature, and that the authors have verified during their real work in hospitals, that is: inadequate presentation format of the data to be

recorded (*Küng et al., 2021*); current procedures must be adapted to regulatory changes; inefficient data capture procedure (*Fan et al., 2022*); incomplete data recording (*Chien et al., 2021*); and poor analysis of the results (*Kapetaneas & Kitsios, 2022*). This is the first novelty of this proposal regarding the state of the art.

Currently, the different technologies used in hospitals do not allow the drug management to be completed adequately, since they only cover specific phases such as preparation or administration in certain hospitalization units. This problem has not been adequately addressed by researchers. Although different studies have proposed improvements in the drug management process in hospitals using disruptive technologies (*Hussien et al., 2021*; *Sharma, Kaur & Singh, 2021*), they have been focused on the analysis in isolation of the usefulness of only one or two of these technologies, without considering the whole drug management process shown in Fig. 1 and the three data levels: data capture, data storage and data exploitation (*Chen, Chen & Yang, 2020*; *Ebrahimzadeh et al., 2021*; *Ko & Woo, 2018*; *Kim et al., 2019*; *Jamil et al., 2019*; *Garcia et al., 2021*).

Therefore, the computer architecture described here will represent an important advance in the state of the art. The proposed architecture (1) uses and combines different disruptive technologies such as blockchain, RFID, QR, internet of things, artificial intelligence and big data; (2) covers the whole drug management process in hospitals; and (3) considers the three data levels (data capture, data storage and data management), to overcome the five existing problems regarding drug management in hospitals. This is the second novelty of this proposal regarding the state of the art.

The third novelty of this proposal regarding the state of the art is the comparative study of the different solutions for labelling and reading data on drug packages (IoT, RFID UHF, RFID NFC and barcodes/QR), considering economic as well as operational and environmental aspects. The results improve the state of the art since existing studies do not compare the four possibilities together (*Liu et al., 2022*; *Ruan et al., 2018*; *Camacho-Cogollo, Bone & Iadanza, 2020*; *Lee et al., 2019*; *Haddara & Staaby, 2018*; *Ebrahimzadeh et al., 2021*; *Vagaš et al., 2019*; *Ko & Woo, 2018*; *Zhang, Fu & Li, 2019*; *Buthelez et al., 2022*; *Cocian, Morales & Schneider, 2023*; *Petro et al., 2009*; *Iadanza, 2012*; *Wu, Kuo & Liu, 2005*; *Bevilacqua et al., 2013*). In contrast, this study has analysed the four technologies and has shown the advantages and disadvantages of Barcode/QR, RFID and IoT for drug management in hospitals. The results are useful to solve three problems related to the management of medicines in hospitals identified in the section "Existing problems with the traceability of medicines in hospitals:" inadequate presentation format of the data to be recorded, current procedures must be adapted to regulatory changes, and inefficient data capture procedure.

The fourth novelty of this proposal regarding the state of the art is the use of the blockchain to assist in the use and management of medicines, from the moment they arrive at the hospital until they are dispensed to patients. Among the existing studies on the use of blockchain for drug management, only one considers activities related to drug storage, management, and dispensing inside hospitals (*Jamil et al., 2019*). The others are focused on other aspects such as the supply chain (*Li et al., 2021*; *Kumiawan, Kim & Ju, 2020*; *Jangir et al., 2019*), mainly with the aim of preventing drug counterfeits (*Pandey & Litoriya, 2021*;

*Uddin et al., 2021*; *Sylim et al., 2018*), information interchange with other hospitals (*Yazdinejad et al., 2020*; *Kim et al., 2019*), the procurement process (*Wang et al., 2021*), patient authentication (*Fan et al., 2018*; *Yazdinejad et al., 2020*), medical history (*Omidian & Omidi, 2022*) or electronic prescription (*Pandey & Litoriya, 2021*; *Garcia et al., 2021*). In addition, existing studies have been conducted on public blockchain networks such as Ethereum (*Tian, 2017*; *Li et al., 2021*; *Jangir et al., 2019*) instead of a private network like the one proposed in this project.

The proposed computer architecture is novel as it is the first to combine the reading of the data with information about the drug and the operations carried out on it, and its registration in a private blockchain to achieve immutable traceability of the drug.

On the other hand, the tendency is to boost the sharing of health data both for research and for better patient care (*EU, 2023*). Data for research is anonymized and aims to improve drug discovery and drug adverse events, for example. Data for better patient care is not anonymized. For example, if hospitals share patient medical records, which include drug dispensations and reactions, patients avoid the need to record this information and take it with them when they go to different hospitals. This is easier to implement in networks of public hospitals where politicians can impose it on all the hospitals, but it looks as though this is the direction that will be taken in the future. The proposed architecture based on a Distributed Ledger Technology like Blockchain could give support to it. Therefore, the traceability of medicines through blockchain will make it possible to solve the following issues related to the management of medicines in hospitals identified in section "Existing problems with the traceability of medicines in hospitals:" Current procedures must be adapted to regulatory changes, and Incomplete data recording.

The fifth novelty of the project regarding the state of the art is the proposal for a third generation Business Intelligence system (*Attar & Chalmeta, 2023*) that takes advantage of the new data that will be stored thanks to the blockchain, and the possibilities offered by big data and artificial intelligence for data processing. The need for Business Intelligence systems in hospital management has been highlighted by different authors, but there are few examples in the context of drug management (*González-Pérez et al., 2022*) and none of them considers both structured and unstructured data from different sources as the proposed architecture does. Existing studies focus on processing structured data obtained from the hospital ERP (*González-Pérez et al., 2022*; *Sajogo, Teoh & Lebedevs, 2023*), sometimes using AI (*Galli et al., 2021*; *Xia, 2022*; *Raza et al., 2022*), or unstructured data from social networks (*Li et al., 2019*; *Hajjami, Berrada & Harti, 2020*).

In addition, the proposal will imply a boost for the use of smart contracts, big data and AI in business intelligence systems, since to date there has been little research on using them for business intelligence purposes (*Hanqing, Mengyue & Jianling, 2022*). Finally, the proposal contributes to making health managers and personnel aware of the possibilities offered by data to help in decision-making, which is one of the limitations that have been identified in the literature for organizations to take advantage of the opportunities it offers (*Mathrani, 2021*). The results will be useful to solve the problem related to the management of medicines in hospitals identified in the section "Existing problems with the traceability of medicines in hospitals:" Poor analysis of the results.

### Social and economic impact

The proposed computer architecture will have a significant impact on the health care of hospital patients and the service quality by reducing medication errors and ensuring that patients receive the correct drug, at the right place and time, and in the desired conditions. This will produce not only health but also economic benefits. These benefits are detailed below.

*Foreseeable health care impact*: It will allow compliance with the current regulations on traceability of medicines in hospitals; It will reduce medication errors through the unique identification of the patient and the professional, and the automatic verification of the drug code; It will ensure the quality and safety of medicines; It will reduce drug-related problems, thereby avoiding the costs associated with solving problems, and will allow the identification of strategic lines to solve medication-related problems; It will improve efficiency in the supply chain, thus ensuring an optimal flow of resources; It will make it possible to identify all the patients who have received a specific drug, vaccine, biological product, prosthesis or medical device; It will facilitate the traceability of how unwanted/ unexpected events have occurred and will allow the detection of misuse of patient information when it is suspected that privacy has been violated; It will allow monitoring of clinical practice data, as well as epidemiological surveillance and clinical research; and It will make it possible to know and monitor the flow of the care process, through the interoperability of clinical information systems.

*Economic impact*: Reduction in the number of telephone calls received in the pharmacy service, as well as those made to check the status of the drugs that require preparation in a hospital pharmacy clean room before being sent for administration to the patient; Cost reduction by eliminating manual records and facilitating software interoperability with electronic medical records; Timesaving in the different stages of the drug process in hospitals; Making better strategic management decisions, based on the analysis of the care activity carried out, the identification of its weak points, the characterization of patients, and the perception of interactions and trends; and Optimization in the allocation of resources.

## LIMITATIONS

It is important to highlight the limitations of the proposal and future research. The proposal presented in this study is conceptual and is based on a review of the literature and the professional and research experience of the authors in informatics and pharmacology. Therefore, future work should be focused on developing and implementing this proposal in different hospitals in order to refine and validate it, as well as to use real data to quantify the expected health care and economic benefits.

## CONCLUSION

The most prevalent types of care errors in hospitals are medication errors. To avoid them, it is necessary to solve the following problems related to the management of medicines in hospitals: the presentation format of the data to be recorded based on barcodes is inadequate; the current procedures must be adapted to the regulatory changes that are taking place in different countries; the procedure currently used to capture the data is

inefficient; not all the data that would be necessary is recorded; and there is no adequate analysis of the results.

To solve these problems, this article has proposed a computer architecture based on different disruptive information technologies such as IoT, blockchain, big data and AI that allow improvements in the process of labelling and reading drug data; the traceability of medicines from the moment they arrive at the hospital until they are dispensed to the patient, thereby ensuring the immutability, quality and security of the data; and the exploitation of data, generating information and knowledge that can be used by hospital managers and health personnel for decision-making in the management and use of medicines.

The computer architecture, once implemented in a hospital, will allow the reduction of medication errors in all phases of the intra-hospital drug process, the automation of work processes, immediate feedback to the personnel involved in the patient's care process, and the improvement of decision-making in the use and management of medicines.

This study has several implications for academics and practitioners. On the one hand, it has been shown that research on the use of IoT and blockchain for drug management in hospitals and drug data exploitation for decision-making is scarce. Therefore, it opens up different challenges for academics, who could develop reference models with best practices about how and when to use IoT, how to develop the blockchain, and what information could be generated to support decision-making. Concerning practical implications, this article shows healthcare executives, managers, hospital administrators, information technology executives, and system analysts the computer architecture that smart hospitals should have in order to improve their performance.

## ABBREVIATIONS
Table 4 shows the abbreviations.

| Table 4 Abbreviations. | |
|---|---|
| **Abbreviations** | **Definition** |
| AI | Artificial intelligence |
| API | Application programming interface |
| ETML | Extraction transformation migration and loading tools |
| EU | European union |
| IoT | Internet of things |
| ME | Medication errors |
| NFC | Near field communication |
| OLAP | On-line analytical processing |
| QR | Quick response code |
| RFID | Radio frequency identification |
| SQL | Structured query language |
| USD | United States dollar |
| UHF | Ultra high frequency |
| VPN | Virtual private network |

### Funding

This work was supported by UJI-FISABIO 2021 under Grant A14 TRAZAMED, by UJI under UJI-B2022-15, and by the Hospital General Universitario de Elche. The funders had no role in study design, data collection and analysis, decision to publish, or preparation of the manuscript.

### Grant Disclosures

The following grant information was disclosed by the authors:
UJI-FISABIO 2021: A14 TRAZAMED.
UJI: UJI-B2022-15.
Hospital General Universitario de Elche.

### Competing Interests

The authors declare that they have no competing interests.

### Author Contributions

- Ricardo Chalmeta conceived and designed the experiments, performed the experiments, analyzed the data, prepared figures and/or tables, authored or reviewed drafts of the article, computer architecture design, and approved the final draft.
- Andres Navarro-Ruiz conceived and designed the experiments, performed the experiments, analyzed the data, authored or reviewed drafts of the article, and approved the final draft.
- Leticia Soriano-Irigaray conceived and designed the experiments, performed the experiments, analyzed the data, authored or reviewed drafts of the article, and approved the final draft.

### Data Availability

  The raw data is available as a Supplemental File.

### Supplemental Information

Supplemental information for this article can be found online at http://dx.doi.org/10.7717/peerj-cs.1455#supplemental-information.

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
