# Peer review of "A computer architecture based on disruptive information technologies for drug management in hospitals"

_PeerJ Computer Science, doi:10.7717/peerj-cs.1455_

## Round 0.1 · original submission · Major Revisions

Dear Authors,
Please revise and resubmit your manuscript. Also, clearly present the contribution of this work to the field. Thank you.

·

Basic reporting

Journal: PeerJ Computer Science
Manuscript Title: A computer architecture based on disruptive information technologies for drug management in hospitals
Manuscript ID: PeerJ-82436v1
Submission Date: Monday, April 3, 2023
The manuscript presented “a computer architecture for drug management in hospitals that is based on the disruptive information technologies”. However, the major and critical weak points are that:
(1) Their proposed work discussion is weak distributed to be described or analyzed.
(2) The novelty is not guaranteed.
(3) Their work is not compared with state-of-the-art approaches nor related studies.
(4) Their experiments leak from the descriptive and statistical analysis.

Experimental design

The rest of my review presents other weak points, comments, and opinions in detail.
Overall Comments:
(1) [KEYWORDS] The keywords (i.e., index terms) should be sorted in alphabetical order.
(2) [ABSTRACT] The abstract should contain the best-achieved results from the performed experiments.
(3) [ABSTRACT] The abstract should reflect the contributions of the manuscript. I suggest rewriting it.
(4) [INTRODUCTION] The authors should provide a clear problem definition and contributions in the introduction section.
(5) [RESEARCH QUESTION] Where is the research question and research gap?
(6) [RESEARCH QUESTION] The research question is not well-formulated or is poorly motivated, and the paper does not provide new insights or information that is not already known.
(7) [RELATED WORK] Where are the related studies? They should be declared in a separate section.
(8) [RELATED WORK] A table of comparisons should be added at the end of the related studies section to praise the pros. and cons. of them. The year column should be added and they should be ordered by it.
(9) Figure 2 is too long.
(10) [METHODOLOGY] The suggested approach is not clearly discussed. More scientific details should be added.
(11) [METHODOLOGY] Where is the overall pseudocode? Flowchart? of the suggested approach?
(12) [Methodology] The study suffers from significant methodological issues that undermine the validity and reliability of the findings.

Validity of the findings

(13) [ABBREVIATIONS] The authors should add a table of abbreviations in the revised manuscript.
(14) [CONCLUSIONS] The conclusions in this manuscript are primitive. Please, write your conclusions.
(15) [REFERENCES] There are no citations for many sentences in the manuscript. Why? Please check.
(16) [REFERENCES] The references should be written in the same style following the journal authors’ guidance.
(17) [REFERENCES] Recent citations from 2021 to 2023 should be added to the manuscript.
(18) [PROOFING] The authors should get editing help from someone with full professional proficiency in English.
(19) [PROOFING] The manuscript should be checked again to fix any typos such as missing spaces and commas.
(20) [CONSISTENCY] The manuscript structure is too short. It must be elaborated in their applied technology as should support more rigorous technical aspects.
(21) [CONSISTENCY] Some paragraphs are wrapped in more than 10 lines. They should be split concisely.
(22) [NOVELTY] What is the novelty of the suggested approach?
(23) [FIGURES] The authors should provide high-resolution figures in the manuscript. For example, Figure 1.
(24) [LIMITATIONS] What are the limitations of the current study? It should be added in a separate section.

Additional comments

For the authors in case of the authors got a chance to review the manuscript and submit the revised one after the editor’s decision, please, provide a table in the revised manuscript mentioning (1) the comment, (2) the authors’ response, and (3) the authors’ change (if applicable). Please, consider all of the comments and don’t ignore any of them. Please, refer to the attached file "PeerJ-82436v1 Reviewer.pdf" for the same comments in an organized format.

Reviewer 2 ·

Basic reporting

In this study, the authors propose a qualitative study of how 3 academics and 3 practitioners feel about a design of RFID + blockchain + business intelligence. The major issue is that the whole blockchain process to record drug flowchart happens within each hospital, which defeats the purpose of using decentralized blockchain. On the other hand, why would each hospital want to exchange their own drug usage information with other hospitals (Figure 2) is unclear.

Experimental design

Totally 6 questions were asked to the 6 experts, resulting in a summarizing table. The findings are reasonable, yet the number of participants are too small to form statistical significance.

Validity of the findings

The agreements among experiments should be calculated.

---

## Round 0.2 · Minor Revisions

Dear Authors,
Please address the minor changes suggested by the reviewer. Thank you.

·

Basic reporting

Journal: PeerJ Computer Science
Manuscript Title: A computer architecture based on disruptive information technologies for drug management in hospitals
Manuscript ID: PeerJ-82436v2
Submission Date: Wednesday, May 17, 2023
The authors have considered most of the raised comment in their updated manuscript. However, there are some comments need to be considered.
(1) Figure 3 is not written in English.
(2) Figure 1 need to be expoerted well as an image as the red line below the text should be removed.
(3) The captions need to be enhanced. More details should be added.
(4) The captions of tables should be added before them not after.
(5) The abstract should be handled as a single paragraph. Remove the new lines between them.
(6) Move the limitations section before the conclusions.

Experimental design

Raised in the first block "Basic reporting".

Validity of the findings

Raised in the first block "Basic reporting".

Additional comments

Raised in the first block "Basic reporting".

Reviewer 2 ·

Basic reporting

no comment

Experimental design

no comment

Validity of the findings

no comment

Additional comments

The authors have addressed my previous comments.

---

## Round 0.3 · accepted · Accept

Dear Authors,

Your article is now Accepted. While in production, kindly check the typos and make the minor English language corrections. Thank you.